# Energy-Related Assessment of a Hemicellulose-First Concept—Debottlenecking of a Hydrothermal Wheat Straw Biorefinery

**DOI:** 10.3390/molecules30030602

**Published:** 2025-01-29

**Authors:** Stanislav Parsin, Marvin Scherzinger, Martin Kaltschmitt

**Affiliations:** Institute of Environmental Technology and Energy Economics (IUE), Hamburg University of Technology (TUHH), Eissendorfer Strasse 40, 21073 Hamburg, Germany

**Keywords:** autohydrolysis, steaming, xylooligosaccharides (XOS), xylan, fractionation, ultrafiltration, modeling

## Abstract

A hemicellulose-first approach can offer advantages for biorefineries utilizing wheat straw as it combines lignocellulose fractionation and potentially higher added value from pentose-based hemicellulose. Therefore, a tailored hydrothermal concept for the production of xylooligosaccharides and xylan was investigated. The focus was on assessing the energy requirements and potential improvements based on experimental results. The wheat straw pretreatment and the downstream processing of hemicellulose hydrolysate were modeled at a scale of 30,000 tons of wheat straw dry mass per year. The results confirmed that the hydrothermal concept can be implemented in an energy-efficient manner without the need for additional auxiliaries, due to targeted process design, heat integration and a high solids loading during hydrolysis. The resulting specific energy requirements for pretreatment and hydrolysate processing are 0.28 kWh/kg and 0.13 kWh/kg of wheat straw dry mass, respectively. Compared to thermal hydrolysate processing alone, the combination of a multi-effect evaporator and pressure-driven ultrafiltration can reduce the heating and cooling energy by 29% and 44%, respectively. However, the ultrafiltration requirements (e.g., electrical energy, membrane area and costs) depend heavily on the properties of the hydrolysate and its interactions with the membrane. This work can contribute to the commercially viable ramp-up of wheat straw multi-product biorefineries.

## 1. Introduction

Biorefineries are already contributing to the defossilization of the industry [1]. First-generation biorefineries, which primarily process food and feed materials, are widely established and produce numerous products from renewable raw materials at an industrial scale under commercial conditions [2,3]. Next-generation biorefineries aim to increasingly utilize lignocellulosic biomass. Agricultural residues in particular, such as wheat straw, are available worldwide, are underutilized and are not in direct competition with the food and animal feed market [1,4].

However, lignocellulosic biomass is recalcitrant and difficult to degrade via biological processes [5,6]. This resistant structure requires at least one pretreatment step, e.g., a subsequent saccharification, in order to achieve sufficiently high product yields [5]. Typical challenges are the high energy requirement (especially in pretreatment and product purification) and the typically complex downstream processes [7,8], i.e., the technical processing of lignocellulosic biomass is often more expensive and complex compared to first-generation biorefineries [1].

As a result, products derived from lignocellulosic materials in such biorefineries have consistently demonstrated a lack of market competitiveness compared to existing options (including first-generation biorefineries) [9,10]. Furthermore, no general concept has yet been established that combines complete feedstock utilization, low operating costs and high yields of products possessing significant market value [1,11].

Wheat straw, as a well-researched example of a lignocellulosic feedstock, has significant potential for the provision of biofuels and bio products [4,12]. As an agricultural residue (sometimes without direct competitive use), it can be a comparatively inexpensive feedstock [4]. Nevertheless, it is still challenging to exploit this potential in biorefineries on an industrial scale under commercial conditions [5,13]. A wide variety of different technologies for the pretreatment of lignocellulosic material, such as wheat straw, have been investigated at least at the laboratory scale (or larger) [14]. One of these pretreatment options moving closer to technological maturity (at least for a continuous operation) is hydrothermal treatment, for example, with steam [15,16]. The application of hydrothermal treatment, such as steam explosion, is usually aimed at maximizing the yield of low molecular weight sugars and the provision of biofuels, e.g., bioethanol or biogas [1,10]. However, processing the resistant lignocellulose exclusively into fuels limits the exergy obtainable from the biofuels [17]. From an economic point of view, co-products with high added value must therefore be provided, for example, from hemicellulose. However, hemicellulose, as the most thermally unstable part of lignocellulose, is usually degraded to compounds of low value and also makes the processing of intermediates more difficult [18,19].

In this context, a hemicellulose-first approach can offer multiple advantages. Analogous to lignin-first concepts [20,21], the approach aims for maximum added value from the hemicellulose fraction in primary refining (i.e., first fractionation step). Such a concept can, for example, provide water-soluble xylans (WSX) and xylooligosaccharides (XOSs), which constitute a significant portion of the wheat straw hemicellulose (i.e., arabinoxylan) [6,22]. The concept of utilizing arabinoxylan to produce short-chain xylans is well-established [23,24]. However, scalable and continuous hydrothermal methods producing competitive products in line with the current requirements of a circular economy (i.e., without resource intensive catalysts, chemical additives and low enzyme loadings) are still not market mature [11,25,26]. Many laboratory-scale processes are unsuitable for large-scale implementation due to technical or economic limitations [1,26]. While conventional steam explosion is used industrially (e.g., for bioethanol production), it is less suitable for the targeted production of biopolymers [1,25]. Moreover, steam explosion is energy-intensive and, therefore, typically expensive [5,17]. In parallel, value-added applications of xylans and xylooligosaccharides (XOSs) in various fields are currently being explored [22,27], i.e., a market for such products is slowly emerging. Thus, a hemicellulose-first approach hypothetically offers both energy and economic advantages, as the hemicellulose fraction is initially solubilized in non-monomeric form during hydrothermal processes [23,28]. Experimentally, relatively high WSX/XOS recoveries have already been achieved with a simple hemicellulose-first concept [29]. The applied combination of adapted steam treatment optimized for a subsequent membrane-based solvent separation can be realized under relatively mild conditions and with low technical complexity. The membrane processes, especially, can help to overcome a typical bottleneck of such lignocellulose biorefineries—the high energy demand in the downstream processing, in particular for thermal energy [7,17,30]. Nevertheless, a major drawback of membrane processes is their relatively high costs, especially for investment and periodic membrane replacement, depending strongly on the respective operation conditions and the properties of the feed solution [31,32]. The energy demand and ultrafiltration requirements must, therefore, be examined in detail based on experimental results.

In this context, this work aims to evaluate a hydrothermal hemicellulose-first concept for the production of xylooligosaccharides (XOSs) and water-soluble xylans (WSXs) from wheat straw based on technical- and energy-related key figures. Since the production of XOS/WSX must be linked to an established processing of cellulose and lignin for holistic feedstock utilization, only the process route for the hemicellulose fraction is examined in more detail. The focus here is on the energy requirement and its reduction without additional auxiliaries, besides water and energy. To achieve this goal, the paper is structured into two parts.

In the first part, a process model is developed based on experimentally determined data for the proposed hemicellulose-first concept scaled to a size of 30,000 t/a (30 kt/a) of wheat straw dry mass [29]. This process approach (reference case) includes a multi-effect evaporator within the (thermal only) downstream processing.In the second part, the downstream processing section is extended to include an ultrafiltration step for the hemicellulose hydrolysate before the evaporator (improved process, alternative case). The impact on product separation and energy requirements is examined in order to be compared with the reference case. Additionally, the equipment cost factors for various investigated membranes are examined, as they need to be weighed against the potential energy savings in the improved downstream processing.

The process models are used to determine the detailed energy and mass balances for the feedstock preparation, hydrothermal hydrolysis and the hemicellulose hydrolysate downstream processing. The results can be used to evaluate the absolute and specific energy requirements of the concept in the context of the experimental results.

## 2. Process Analysis (Reference Case)

This section describes the procedure for modeling and analyzing the reference case (conventional process).

### 2.1. Process Modeling—Reference Case

In the following, the modeling approach is described in more detail. First, a process definition is given and the underlying process assumptions are listed. Subsequently, the simulation tools are presented.

#### 2.1.1. Process Definition—Reference Case

For the modeling, a wheat straw biorefinery concept was developed based on a hemicellulose-first approach. Such an approach prioritizes high added value from the hemicellulose fraction (referred as arabinoxylan). In this case, the proposed first fractionation step (primary hydrolysis) of wheat straw with saturated steam (without explosion) is applied to realize the highest possible arabinoxylan recoveries as xylooligosaccharides (XOSs) and water-soluble xylan (WSX) in the hemicellulose hydrolysate. The approach enables the complete utilization of the raw material wheat straw, whereby the cellulose and lignin fraction should remain as completely as possible in the solid residue and should not contaminate the hemicellulose hydrolysate [29]. Here, however, only the processing of the hemicellulose (i.e., arabinoxylan) faction is examined in detail.

The overall modeled process is shown in a simplified form in Figure 1. For continuous operation, the changes described below have been adopted. Figure 1 represents the reference case, which implies that the hydrolysate downstream processing only involves (thermal) thickening by means of a multi-effect evaporator (MEE). The multi-effect evaporator is modeled as a three-stage flash system. In the overall process, three areas are differentiated (preparation, hydrolysis and downstream processing) in order to allocate heat and cooling demand for analysis and heat integration.

The composition of the XOSs/WSX hydrolysate, which is produced during primary hydrolysis and is to be thickened to the target concentration during downstream processing, is given in Table 1 (further details are given in the Appendix A).

The following assumptions were made for modeling the process on an industrially useful scale.

A continuous process in steady state with 7500 h/a (full load hours per year) was assumed.The process is scaled to a comparable size of 30 kt_DM_/a of wheat straw as feed corresponding to a feed rate of 4444 kg/h of wheat straw (initial 90 wt% dry mass and 45 wt% after preparation in Figure 1). This scale was chosen with regard to existing plants [1].Heat was provided as high-pressure steam at 250 °C and low-pressure steam at 125 °C.Cooling water was supplied at 20 °C and heated to a maximum of 25 °C.The efficiencies of the pumps and the drivers were assumed as 80% and as 90%, respectively [33].The electrical energy demand for raw material preparation was assumed to be 15 kWh/t for a bale shredder and 15 kWh/t for a cutting mill and the respective periphery [34,35]. This is a very conservative assumption [36], as a stalk length of above 1 cm is defined as the target value [29].For the preparation, the temperature of the soaking/washing water was assumed to be 90 °C.For the extraction, a continuous countercurrent-suspension extraction with a yield of 94% for the target fraction (XOS/WSX) was assumed [37].Further processing of the solid lignin and the cellulose (solid residue in Figure 1) was not considered as both fractions can be used for established applications after hydrothermal pretreatment.A dry mass content of 10% and the analytically determined dry mass composition (Table 1) was assumed for the hydrolysate to account for optimized primary hydrolysis and effective continuous extraction on an industrial scale [15,37].The multi-effect evaporator (MEE) was designed as a three-stage flash at 1 bar, 0.4 bar and 0.2 bar. The first two condenser stages of the three-stage (multi-effect) evaporator were not considered in heat integration, as they supply the next stage with heat.The final dry mass content in the concentrate after the multi-effect evaporator was 50%.

#### 2.1.2. Flowsheet Simulation—Reference Case

The concept, which was modeled with a corresponding focus on energy requirements, is shown in Figure 1. The processes were modeled in Aspen Plus^®^ (V9, Aspen Technology, Bedford, MA, USA) to provide the mass and energy balances and to gain insights into the given possibilities of heat integration. All process parts were simulated as stationary processes. Mechanical operations were implemented as short-cut models based on empirical data (steady-state models). Thermal operations were simulated rigorously (time independent equilibrium models). The NRTL (Non-Random Two Liquid) method for strongly non-linear systems was used to model the vapor–liquid and liquid–liquid equilibria. Here, the equilibria for a binary system were determined using the activity coefficients of the substance *i* or *j* ($\gamma_{i/j}$) as a function of their overall mole fractions ($x_{i/j}$) according to Equation (1). The excess enthalpies ($G_{ij/ji}$) were calculated according to Equation (2). The coefficient *τ* was determined from the energies of the interactions (g) between an *i-j* pair of molecules, the gas constant (*R*) and the absolute temperature (*T*) according to Equation (3). The coefficient *τ* and the non-randomness constant for binary interactions *α* (between 0.2 and 0.47) were determined experimentally according to [38] and retrieved from the Dortmund Data Bank [39].(1)$${\ln\gamma }_{i/j}=x_{j/i}^{2}\left[\tau_{ji/ij}{\left(\frac{G_{ji/ij}}{x_{i/j}+x_{j/i}G_{ji/ij}}\right)}^{2}+\frac{\tau_{ij/ji}G_{ij/ji}}{{\left(x_{j/i}+x_{i/j}G_{ij/ji}\right)}^{2}}\right]$$(2)$$G_{ij/ji}=e^{\left(-\alpha_{ij/ij}\tau_{ij/ji}\right)}$$(3)$$\tau_{ij/ji}=(g_{ij/ji}-g_{jj/ii})/RT$$

The components and properties were defined and customized following a guideline provided by NREL (National Renewable Energy Laboratory) [40]. All common pure substances were defined as conventional components. Values for physical and phase properties were taken from the APV90 database (i.e., PURE32, AQUEOUS, SOLIDS, INORGANICS). All insoluble components, such as cellulose, lignin, xylan, proteins, ash, etc., were defined as solids. All missing thermodynamic or transport parameters for the calculations were either approximated using the chemical formula or added according to literature [40]. For the modeling of chemical reactions, the RSTOIC model was used with the empirically determined conversion rates. Aspen Energy Analyzer^®^ (V9, Aspen Technology, Bedford, MA, USA) was used for pinch analysis and heat integration considering the assumptions listed in Section 2.1.1.

### 2.2. Results and Discussion—Reference Case

Below, the results of the modeling and analysis for the conventional process (reference case) are presented and discussed. The process models and calculations were based on experimental data and analytical examinations according to [29].

**Results.** Under the assumptions for a plant of 30 kt_DM_/a wheat straw stated above, about 7250 kg/h of hydrolysate with 10 wt% dry mass and a composition according to Table 1 was obtained. Figure 1 shows the conventional process with the multi-effect evaporator in downstream processing to thicken the hydrolysate to 50 wt% dry mass.

The energy balances for this approach are shown in Figure 2A–C. The respective heating (Figure 2A) and cooling requirements (Figure 2B) were differentiated according to the causal process areas (Figure 1). These two diagrams show the heating and cooling requirements in the respective process areas before integration. Additionally, Figure 2C shows the final installed capacities required after heat integration. Thus, the entire heat demand for the preparation (483 kW in Figure 2A) and additional 193 kW in the hydrolysate downstream processing can be covered by heat integration. In both cases (i.e., 676 kW), the integrated heat demand can be covered by condensing the excess steam after primary hydrolysis (Figure 1). This leads to a corresponding reduction of 675 kW (rounding deviation) in cooling demand in the hydrolysis area (Figure 2B). In total, 22.4% (3024 kW in Figure 2A vs. 2348 kW in Figure 2C) of heating and 26.5% (2547 kW in Figure 2B vs. 1872 kW in Figure 2C) of cooling energy can be covered by integration.

In the conventional process (reference case), most of the resulting energy (52% of heat and 89% of cold after heat integration) is required in the downstream processing of the hydrolysate. The final requirements are listed in Table 2 for the process example considered (30 kt/a wheat straw dry mass).

In Table 2, the reference case is characterized by the required inputs and the possible product outputs after heat integration according to Section 2.1 and Figure 2C. In the pretreatment (i.e., feed preparation and primary hydrolysis), the specific heat demand per ton of wheat straw dry mass is 0.28 MWh (i.e., 8453 MWh/a for 30 kt/a wheat straw dry mass). For the downstream processing of the provided hydrolysate, the specific heat demand is 0.31 MWh/t wheat straw dry mass (i.e., 9157 MWh/a for 30 kt/a wheat straw dry mass), which is higher than for hydrothermal primary hydrolysis. High-temperature steam is required for primary hydrolysis at 180 °C. The electricity demand is relatively low at 1020 MWh/a (i.e., 136 kW; Figure 2C). The composition and amount of solid residues, the total amount of XOS/WSX concentrate and the share of the target fraction within the concentrate mass flow are shown in Table 2. Under the assumed conditions, about 10.3% of the wheat straw biomass (dry) is obtained as non-monomeric xylose (i.e., XOS/WSX) in the concentrate and another 79.6% as solids (cellulose and lignin).

**Discussion.** A major challenge in implementing lignocellulose-based biorefineries is the typically high operating costs, which are mainly due to high energy requirements and process aids that hinder the effective use of economies of scale [15,41]. Thus, the preceding approach was designed with respect to recommendations in the literature (i.e., simplest possible design for continuous implementation with high solid loadings and without expensive auxiliary demands) [13,30]. Auxiliary materials, such as acids, bases, enzymes or adsorbents, are often resource-intensive in their production and are, therefore, characterized either by high production costs/market prices or by considerable environmental impacts, which is not in line with the objectives of the bio-based circular economy [13]. Additionally, for the steam-supported hydrolysis, no energy-intensive raw material preparation (e.g., fine milling, pelletizing) resulting in a considerable specific energy consumption is needed [8,35]. Soaking of the straw at 90 °C (Figure 1) is suggested as it removes impurities from the raw material without requiring additional energy and without affecting the target fraction (i.e., lignocellulose). Thus, the straw is additionally preheated for the subsequent primary hydrolysis without additional energy demand; heat at 90 °C can effectively be provided by heat integration (i.e., condensation with at least 10 K temperature difference). Also, the preparation with water does not generate wastewater requiring special treatment, unless the straw has been exposed to harmful chemicals. In this approach, the straw is only dedusted, coarsely chopped and not treated by organic solvents, acids or bases, i.e., no extensive wastewater treatment is necessary.

The primary hydrolysis has a relatively low thermal energy requirement, since no large quantities of solvents or auxiliary materials (as for liquid hot water or organosolv pretreatment) have to be heated (i.e., high solids loading) [8,42]. Only the preheated raw material with its water content is heated. The experimental data show that the water content of the straw must be above 50 wt% for a sufficiently high XOSs/WSX yield [29]. Therefore, a water content of the straw after preparation of 55 wt% is assumed within the process model resulting in a liquid-to-solid ratio of about 1.2 for hydrothermal hydrolysis; this value is considerably lower than usually required for autohydrolysis/liquid hot water treatment [15,16,28].

The treated (hot) biomass/straw also provides thermal energy for extraction, being more effective at higher temperatures [43,44]. A high extraction yield for the target fraction XOS/WSX with the lowest possible use of water (solvent) is important, as otherwise more extraction stages or a higher liquid-to-solid ratio must be applied (Section 2.1.1); the latter again increases the heat demand in downstream processing due to dilution of the product.

The concept energy demand is assessed based on the assumption that excess steam can be separated in a flash chamber at a pressure of ca. 1 bar according to Figure 1 (Section 2.1.1). In this context, the accumulation of volatile components, such as acetic acid and furfural in the vapor phase, must also be taken into account [29]; however, based on existing process engineering approaches, solutions should be available. Whether the concept and this type of vapor recovery can also be implemented continuously without technical problems still needs to be validated experimentally.

Another factor of concern is the reactor design. In order to maintain the necessary retention time by realizing the required throughput, the reactor needs to be designed correspondingly large [40]. Since a screw reactor is proposed and the temperature as well as the working pressure are relatively low in the context of a hydrothermal treatment, and many examples of large-scale screw conveyers are already available [45,46], adequate technical solutions are on hand.

In summary, the results show that the proposed approach is a comparatively simple and effective way to fractionate wheat straw lignocellulose and to isolate potentially valuable XOS/WSX from the hemicellulose (referred as arabinoxylan). No auxiliary materials are required in addition to energy and water. Besides this, no energy-intensive steps are required during feedstock preparation and the thermal energy applied can be used in an integrated way. For a low energy requirement, it was useful to keep the liquid-to-solid ratio as low as possible during hydrothermal hydrolysis and to utilize the heat energy contained in the (solid and liquid) intermediate streams as effectively as possible (and not simply cool it away).

However, the thermal hydrolysate downstream processing shows a typical significant disadvantage [28]. The evaporator-based downstream processing (Figure 1) is characterized by a higher specific as well as absolute heat and cooling requirement compared to hydrothermal hydrolysis, as the hydrolysate consists mainly of the solvent water making a treatment using only a multi-effect evaporator energy-intensive (Figure 2); thus, this offers a considerable potential for improvement.

## 3. Process Improvement (Alternative Case)

In the following, the investigation of the improved process (alternative case) is described. Therefore, an extension of the process within the downstream processing by including ultrafiltration is investigated. The energy requirements and the equipment costs for the membrane modules are examined in detail, as they must be compared with the potential energy savings. They are also directly dependent on the hydrolysate filtration properties and, therefore, provide a quantifiable criterion of the adapted saturated steam pretreatment [29]. Thus, below, the methodology for the experimental investigations, the process modeling, the cost analysis and the corresponding results are described.

### 3.1. Experimental Procedure—Alternative Case

About 3 kg of prepared wet (37% dry mass) wheat straw (*Triticum* sp.) was treated with saturated steam in a 40 L fixed-bed reactor without any further auxiliaries in order to solubilize especially the hemicellulose fraction. The hemicellulose (referred to as arabinoxylan) is subsequently present predominantly as xylose-based oligomers and polymers (XOS/WSX) and can be separated from the remaining solid residue (mainly cellulose and lignin) as a hydrolysate (composition in Table 1). The pretreatment is aimed to provide a particle- and lignin-poor hydrolysate (approx. 50 to 100 NTU after steaming [29]) to serve as a feed for various membranes. The screening is aimed to identify the most effective enrichment and separation option for the target fraction XOS/WSX (i.e., non-monomeric xylose) using ultrafiltration in the dead-end mode. A triple-batch set-up with membranes in Table 3 (45 cm^2^ membrane area) at 20 to 22 °C was used for ultrafiltration. A transmembrane pressure (TMP) of 1 bar (for UH50) and 4 bar (for all other membranes) was applied.

Figure 3 shows a scheme of the experimental procedure (for further details see [29]).

### 3.2. Process Modeling—Alternative Case

The modeling procedure is analogous to the reference case (Section 2.1). Thus, only changes in the process are described below.

#### 3.2.1. Process Definition—Alternative Case

To evaluate the influence of ultrafiltration prior to the multi-effect evaporator (MEE) on the energy demand, the downstream processing of the conventional process (Figure 1) was modified and modeled according to Figure 4; i.e., the modification of the process includes ultrafiltration of the hydrolysate until a conversion factor (Equation (7)) of 50% is reached.

#### 3.2.2. Flowsheet Simulation—Alternative Case

Ultrafiltration was simulated using the component separator block based on split fractions in Aspen Plus^®^. The distribution coefficients of all relevant feed components in permeate and retentate were analytically determined (Section 3.1).

### 3.3. Cost Analysis—Alternative Case

The equipment costs for ultrafiltration modules were determined reflecting the differences of the investigated membranes. Therefore, only the equipment costs were used; other costs were assumed to depend linearly on these equipment costs and, thus, are not considered here. For the cost estimation, based on the experimental procedure and the setup of ultrafiltration [29], the permeate flux ($\dot{J}$ in L/(m^2^ h)) of a membrane for a certain period (Δ*t* in h) is determined gravimetrically using the permeate volume difference (Δ*V^Per^* in L) and the active membrane area (*A_m_* in m^2^) according to Equation (4). The permeate density was assumed to be 1 kg/L.(4)$$\dot{J}=\frac{\Delta V^{Per}}{A_{m}\;\Delta t}$$

The flux is divided by the transmembrane pressure (*TMP*) in order to compare the membrane performance at different operating pressures. Thus, the permeability ($\dot{P}$ in L/(m^2^ h bar)) is calculated according to Equation (5).(5)$$\dot{P}=\frac{\Delta V^{Per}}{A_{m}\;\Delta t\;TMP}$$

The average permeability ($\bar{P}$ in L/(m^2^ h bar)) during filtration of the sample for each membrane (at constant time intervals (Δ*t*)) is calculated according to Equation (6).(6)$$\bar{P}=\frac{1}{n}\sum_{i=1}^{n}{\dot{P}}_{i}$$

The average permeability accounts for the decreasing value over the conversion factor (*CF* in%) according to Equation (7). The filtration is stopped as soon as the target value for the conversion factor of 50% is reached (i.e., the ratio of the liquids feed (*m^Feed^* in g) to permeate (*m^Per^* in g)).(7)$$CF=\frac{m^{Per}}{m^{Feed}}\;100\%$$

For each filtration, the yield (*Y* in%) of the target fraction (i.e., XOS/WSX) in the retentate (Figure 4) is calculated according to Equation (8) with the concentrations of non-monomeric xylose in the feed ($c_{XOS/WSX}^{Feed}$ in g/L) and in the retentate ($c_{XOS/WSX}^{Ret}$ in g/L) and the respective mass (*m* in g), since the non-monomeric xylose is representative of the XOS/WSX content [29].(8)$$Y_{XOS/WSX}=\frac{c_{XOS/WSX}^{Ret}\;m_{Ret}}{c_{XOS/WSX}^{Feed}\;m_{Feed}}\;100\%$$

The specific and scalable membrane area ($A_{m}^{sp}$ in m^2^ bar) for each membrane necessary to filter the simulated hydrolysate stream (${\dot{V}}_{hyd}^{sim}$ in L/h) is calculated according to Equation (9).(9)$$A_{m}^{sp}=\frac{{\dot{V}}_{hyd}^{sim}}{\bar{P}}$$

Ultrafiltration typically runs at transmembrane pressures of 1 to 10 bar (rarely up to 20 bar; Table 3) and depends on various process conditions [29]. Therefore, a transmembrane pressure (*TMP*)-related cost factor (in EUR bar) according to Equations (10) and (11) is determined depending on the specific membrane area ($A_{m}^{sp}$) required (i.e., equivalent to the costs at a transmembrane pressure of 1 bar).(10)$$TMP\;related\;cost\;factor\;\left(invest\right)=\left(p_{h}+p_{m}\right)\;A_{m}^{sp}$$(11)$$TMP\;related\;cost\;factor\;\left(periodic\right)=p_{m}\;A_{m}^{sp}$$

For the membrane housings and spacers, EUR 90/m^2^ (*p_h_*), and for the membranes EUR 50/m^2^ (*p_m_*) were assumed as average prices according to information from the industry and the literature for the base year 2023 [31]. For the initial investment (Equation (10)), housing, spacer and membranes are provided. With regard to periodic renewals (Equation (11)), only the membranes are taken into account. For the (purchase) equipment costs, the given transmembrane pressure-related cost factors have to be divided by the applied transmembrane pressure (*TMP*), since this can vary in a relatively wide range.

### 3.4. Results and Discussion—Alternative Case

In the following, the results of the improved process (alternative case) are presented and discussed.

#### 3.4.1. Experimental Data

**Results.** The experimental investigation of ultrafiltration is primarily aimed at evaluating the membranes for the effective separation of the target fraction (XOS/WSX) (characterized in Appendix A). The dynamic progression of permeability during ultrafiltration of hemicellulose hydrolysate and the determined average permeability are shown to be exemplarily for the membranes UH50, UH30, UP20, UP10, UP5 and UH4 in Figure 5A–F. In all cases, a typical reduction in the permeability over the increasing conversion factor (*CF*; Equation (7)) can be observed. The reduction in permeability is degressive and appears to be approaching a limit with an increasing conversion factor. The results show several trends.

The smaller the pore size (Molecular Weight Cut-Off, MWCO), the lower the absolute reduction in permeability (i.e., difference in permeability at *CF* = 0% and *CF* = 50%).The relative reduction is larger for the hydrophobic UF and UP series than for the hydrophilized UH membranes (Table 3). In particular, the results of UH30 (Figure 5B) and UH4 (Figure 5F) show a significantly reduced fouling potential of the hydrolysate after saturated steam treatment.The permeability compared to deionized water is significantly reduced from the beginning in all cases investigated and falls sharply in the initial phase of filtration (up to a conversion factor of roughly *CF* = 20%).The regressions follow a second-order polynomial.

**Discussion.** Based on the assumptions that the initial investment costs (housing, spacers and membranes) and the periodic costs for membrane renewal depend linearly on the required membrane area (Equation (9)), the empirically determined average permeability (*AVG* in Figure 5) is decisive. This permeability was determined during the filtration of the produced hydrolysate samples (Table 1) in dead-end mode (i.e., batch mode). By means of these average values, the calculated necessary membrane areas are conservative estimates, since the values were normalized to the number of measuring points over the conversion factor (Section 3.1). In a continuous cross-flow process, the permeability can be higher and might be influenced by the cross-flow velocity [48,49]. However, the specific volumetric energy consumption can be (considerably) higher as a result [50]. For the ultrafiltration of complex media, it is important to determine the necessary membrane area as closely as possible to the applied conditions due to the multiple influencing factors [29]. In the investigated case, cross-flow filtration can prove to be beneficial, as the feed medium has a low dry mass content and is not thickened significantly during ultrafiltration (i.e., the viscosity is not greatly increased).

Since the specific membrane area is proportional to the amount of feed (hydrolysate) according to Equation (9), the required membrane areas and costs can be evaluated for the modeled plant capacity, assuming otherwise constant conditions. Parameters influencing the average permeability (and, thus, the costs) are, for example, the conversion factor (*CF*) and the filtration temperature [31,51,52]. Here, a *CF* of 50% was applied, since it offers an optimal comparability to the feed as reference and provides reasonable conditions for the study (Section 3.1). A higher *CF* (for a higher concentrated retentate) would tend to lower the average permeability (Figure 5). At the applied *CF* of 50%, the decrease in permeability seems to be mainly due to the fouling mechanisms at the beginning (mainly up to 20% *CF*). However, the overall permeability reduction for hydrophilic membranes like UH30 and UH4 is low compared with similar hydrophobic membranes (Appendix A) [29]. The increasing concentration of retained compounds subsequently causes only an insignificant additional reduction. The permeability values (Figure 5A–F) can be extrapolated by means of the regression curves. Tests have shown that the filtration of the hydrolysate up to a *CF* of 80% is possible even with small pore sizes of 4 kDa in dead-end mode. However, the viscosity of the hydrolysate is greatly increased from a dry mass content of more than 20 wt%.

#### 3.4.2. Process Data

**Results.** The modeling of the improved process at a plant size of 30 kt/a wheat straw dry mass shows that the provision of ca. 7250 kg/h of hydrolysate (analogous to the reference case) with 10 wt% dry mass is feasible. In the alternative case, 579 kW of heating in Figure 6A and 580 kW of cooling in Figure 6B (rounding deviation) can be covered by heat integration (Figure 6C), as the heat and cooling demand in downstream processing is significantly reduced compared to the reference case (Figure 2). The introduction of ultrafiltration (at a conversion factor *CF* = 50% according to Equation (7)) significantly changes the absolute energy demand and the energy demand profile (Figure 6). With only a slight increase in the demand for electrical energy for the TMP-pump (about 1 kW_el_ at 4 bar transmembrane pressure (*TMP*)), the heat demand can be reduced by approx. 29% and the cold demand by approx. 44% (Figure 6C) compared to the reference case (Figure 2C). The absolute and relative heating and cooling requirements for hydrolysate downstream processing have been significantly reduced (Figure 2 and Figure 6). Compared to the reference case, the heat requirement of the hydrothermal hydrolysis here accounts for around half of the total requirement (1123 kW Figure 6A) and rises to 68% after heat integration (Table 4).

Table 4 compares the two processes in terms of their energy requirements and product outputs. The results show that the implementation of an ultrafiltration step mainly affects the demand for thermal energy in the form of low-pressure steam, the cooling demand and the product output (XOS/WSX concentrate). With the improved process, 5167 MWh/a of heat and 6116 MWh/a of cold can be saved compared to the reference case. Since the pretreatment (i.e., feed preparation and primary hydrolysis) is identical, the specific heat demand per ton of wheat straw dry mass is 0.28 MWh after heat integration in both cases. For the downstream processing of the hydrolysate, the specific heat demand is reduced by 58% from 0.31 MWh/t (Table 2) to 0.13 MWh/t wheat straw dry mass for the improved process (Table 4). The composition and amount of the solid residues remain unchanged. The total amount of XOS/WSX concentrate is about 24% lower for the improved process (alternative case) compared to the reference case, but the share of the target fraction within the provided mass flow is considerably higher, at 71%_DM_. A permeate stream is produced during ultrafiltration (Figure 4) (at a molecular weight cut-off of 4 kDa) consisting of about 95 wt% water [29].

**Discussion.** If the primary hydrolysis is designed for a subsequent ultrafiltration [29], the process (alternative case) is characterized by a significantly lower thermal energy demand compared to the reference case. Despite the conservative assumptions (e.g., low filtration temperature, low transmembrane pressure and conversion factor of 50%), 5167 MWh/a of thermal energy can be saved in comparison to the reference case (Table 4). Assuming an energy price of EUR 0.06/kWh (EU average, first semester 2021 to 2023), this corresponds to savings of EUR 310,000 per year for low-pressure steam only [53]. Using ultrafiltration, the dry mass content of the hydrolysate can be increased from 10% to 14.7% in the retentate (for UH4) with a conversion factor *CF* = 50% reducing the amount of water to be evaporated in the multi-effect evaporator by over 50%. Mechanical thickening processes are therefore particularly effective for fluids with a low dry mass content, especially in cross-flow mode [48,49]. However, filtration in cross-flow mode has a higher specific energy consumption. The installed power required for filtration in cross-flow mode could increase from around 1 kW_el_ up to 11 kW_el_ [50]. The conversion factor can be further increased, if necessary, which would additionally reduce the energy requirement for the subsequent multi-effect evaporator. However, a high concentration of polymers changes the viscosity and filtration properties of the hydrolysate [32]. A new average permeability (Equation (6)) must then be estimated by extrapolation with the data in Figure 5. However, the retentate must remain pumpable, since the multi-effect evaporator is necessary not only for further thickening but also for removing the volatile components (especially acetic acid and furfural; Table 1).

The considerable amount of thermal energy required for highly diluted hydrolysates can be reduced by using pressure-driven processes such as ultrafiltration (Table 4). However, these processes are only purposeful if the target fraction differs significantly in particle size or molar mass from the interfering materials or bulk. For oligomeric hemicellulose, however, such an approach is well suited and can be implemented effectively if the feed solution shows tolerable fouling properties such as the wheat straw hydrolysate produced in the proposed concept [29].

#### 3.4.3. Cost Evaluation

In the following section, the evaluation results using the parameters defined in Section 3.3 are presented in Table 5 and Figure 7 and then discussed.

**Results.** The specific membrane areas required, the xylooligosaccharides (XOS) and water-soluble (WSX) concentrations and the calculated average permeabilities (Equation (6)) are listed in Table 5. Under the assumed conditions, about 3625 L/h each of permeate and retentate with the specified XOS/WSX concentrations are provided for further processing. The specific membrane areas required differ considerably due to the different average permeabilities and fouling behavior during hydrolysate filtration and vary between 66 and 8840 m^2^ bar.

The empirical data and simulated results were used to estimate the equipment cost factors of ultrafiltration modules for the investigated membranes in Table 3 according to Section 3.3. Therefore, the determined costs are shown as a transmembrane pressure related cost factors in Figure 7; i.e., the cost factor of each membrane is given in EUR bar (Equations (10) and (11)) and must be divided by the respective transmembrane pressure to be applied.

The initial investment cost factors for the equipment of the ultrafiltration plant for 30 kt_DM_/a of wheat straw vary between EUR 9000 bar for the UH50 and EUR 1,238,000 bar for the UF5. The periodic cost of membrane replacement (typically one to two years lifetime) is constant at 35.7% of the investment (Section 3.3). The corresponding ultrafiltration yield for the target fraction XOS/WSX (i.e., non-monomeric xylose) in the respective retentate is shown in Figure 7. The maximum yield is 89% for the UP5 and UF5 membranes, and 88% for the UH4 membrane. Therefore, with approximately the equal yield, the investment and the periodic costs for the hydrophilized UH4 membrane are 57% of the costs for the UF5 and 73% of the costs for the UP5 membrane.

**Discussion.** Significant differences in the required membrane areas and cost factors were found (Figure 7). The ultrafiltration results show that effective enrichment and separation of XOS/WSX is possible at a comparatively low cost, even with low specific energy consumption (Table 4) and without prior extraction, adsorption or precipitation of lignin or other impurities [31,54,55]. The lowest costs with comparatively high yields for XOS/WSX can be achieved with hydrophilic membranes (Figure 7), presumably due to the interaction with the hydrolysate [29]. The wheat straw washing and the adapted primary hydrolysis probably remove foulants and prevent a high solubilization of cellulose and lignin, resulting in a hydrolysate with low turbidity and lignin content (Table 1). These properties, in combination with hydrophilic membranes, lead to sufficient permeabilities even with direct ultrafiltration at moderate filtration pressures and temperatures (Table 5). In addition to the cost factors in Figure 7, further costs (e.g., peripherals, insurance, engineering and maintenance) must also be taken into account for a complete economic evaluation [51,52]. However, the pure purchase costs for ultrafiltration are typically dominated by the module and membrane costs [51,56].

The results suggest that the hydrophilic 50 kDa membrane can be used for prefiltration (Figure 3) at a transmembrane pressure of only 1 bar a filtration without losses of the target fraction at a conversion factor of 100% [29]. Due to the high permeability for the hydrolysate, the required membrane area is relatively small (Table 5) and the costs are comparatively low (Figure 7). Therefore, all coarse impurities greater than 50 kDa can be separated and utilized.

For the enrichment of the target fraction XOS/WSX, a hydrophilized tight membrane like UH4 is suitable, as it shows a very high yield and potentially the longest lifetime at a moderate cost (Figure 7). The lifetime of the membrane modules depends mainly on their susceptibility to fouling and is, therefore, probably the longest for the hydrophilized UH series [52,57], as these membranes show a significantly reduced (irreversible) fouling potential in comparison with the hydrophobic types (UF, PS, PES; Table 3 and Appendix A) [29]. This would have a significant positive impact on the investment and operating costs of ultrafiltration [51,56], while a lower molecular weight cut-off (<4 kDa) increases the yield even further (Appendix A).

Some further factors have a proportional effect on the permeability (e.g., temperature, transmembrane pressure, pH-value) and can be used to influence the determined permeability values [48,49]. The ultrafiltration temperature, especially, is an effective parameter (with an almost linear relationship to the permeability [31]) that can influence the required membrane area and achieve or even reduce the determined costs in Figure 7. According to Figure 1, the hydrolysate will tend to have a higher temperature after hydrothermal pretreatment and extraction in the continuous process (Figure 4) compared to the discontinuous experimental setup described in Section 3.1 (i.e., the hydrolysate does not cool down before filtration). Consequently, higher flow rates in continuous mode at a large scale can be assumed. Therefore, the calculated necessary membrane areas (Table 5) and, thus, the costs shown (Figure 7) tend to be a conservative estimate (overestimation).

For exemplary ultrafiltration with the UH4 membrane in the improved process (Figure 4) (*TMP* = 4 bar and *T* = 20 °C), about EUR 178,000 as an investment and EUR 63,000 as periodic equipment costs are estimated. This is offset by the savings in heating (approx. EUR 310,000/a under the assumed conditions in Europe) and cooling energy (Table 4).

## 4. Overall Discussion

This study evaluated a hydrothermal hemicellulose-first concept for wheat straw biorefineries, focusing on the hemicellulose valorization. A more effective use of hemicellulose, such as producing xylooligosaccharides (XOSs) and water-soluble xylan (WSX), offers significant opportunities for higher-value applications compared to, e.g., xylose molasses. Notably, while molasses typically sells for $0.1 to $0.3 per kg, XOSs can command prices ranging from $10 to $30 per kg [27,58]. However, for practical implementation in biorefineries, the results must be considered in the broader context of overall lignocellulose fractionation and valorization. The concept investigated here demonstrates energy-efficient production of XOS/WSX concentrates (characterized in Table 4 and Appendix A) under optimized conditions (Section 3.4.2). The integration of the concept into existing hydrothermal biorefineries can contribute to the overall economic viability of multi-product wheat straw biorefineries.

A major part of the arabinoxylan (63.1 ± 3.1%) in the feedstock is solubilized during primary hydrolysis and is mainly present as non-monomeric xylose (about 87%) (Table 1).The majority of the cellulose (over 90%) and lignin (over 70%) remain in the solid residue, which can be utilized for other applications [29].The specific (0.28 MWh/t_DM_ (1 GJ/t_DM_) wheat straw) (Section 3.4.2) energy requirement for feedstock preparation and primary hydrolysis is relatively low [8,16], as no energy intensive preparation is necessary and only the arabinoxylan needs to be solubilized. A low liquid-to-solid ratio, the recovery of excess steam and effective heat integration further reduce the energy requirements (Table 2 and Table 4).If the hydrolysate properties enable effective ultrafiltration, substantial amounts of thermal energy can be saved compared to exclusively thermal downstream processing, with reductions of 29% in heating and 44% in cooling energy (Figure 2 and Figure 6). This results in a comparatively low specific energy requirement for downstream processing of 0.13 MWh/t_DM_ (468 MJ/t_DM_) of wheat straw.The low turbidity of the hydrolysate (approx. 50 to 100 NTU after steaming) indicates a low particle load and lignin content. This leads to a low fouling potential, higher flow rates, reduced membrane area requirements and, finally, lower module and membrane costs, even under moderate filtration conditions (Section 3.4).

The relatively simple process design (Figure 1), along with the avoidance of auxiliaries (besides water and energy) and the confirmed biodegradability of the solid residue (Appendix A), support potential integration into existing concepts for establishing a multi-product biorefinery with reduced waste generation. However, the feasibility of continuous primary hydrolysis must still be experimentally validated, since continuous steam blasting and depressurization equipment may not align with current state-of-the-art technology.

## 5. Conclusions

Xylooligosaccharides and water-soluble xylan can be produced energy efficiently in a hydrothermal wheat straw biorefinery using only water and energy as auxiliary inputs. A hemicellulose-first approach is well suited for implementing such a biorefinery, as the focus on hemicellulose, which is more easily solubilized hydrothermally, combines higher added value with a simplified process design and low specific energy consumption. The results show that for hydrothermal hydrolysis, it may also be advantageous to avoid additional auxiliaries in other process steps in favor of hydrothermal processes, as this enables heat integration without generating additional waste streams. Significant heating and cooling energy savings can be achieved through pressure-driven downstream processing and solvent separation. The energy and cost benefits are enhanced when primary hydrolysis is adapted towards ultrafiltration and the interactions between hydrolysate and membranes are considered. An energy-efficient implementation of a hemicellulose-first approach is achievable by avoiding intensive size reduction, using mild reaction conditions and enabling effective heat integration.

## Figures and Tables

**Figure 1 molecules-30-00602-f001:**
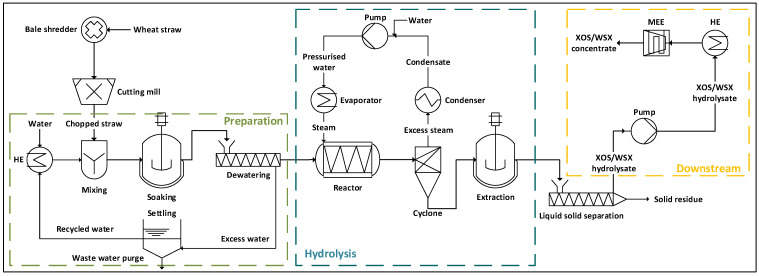
Simplified flowsheet of the reference case process. Only the relevant heat exchangers (HEs) for conventional hydrolysate treatment (XOSs—xylooligosaccharides and WSX—water-soluble xylan) are indicated. In the conventional downstream processing, only a multi-effect evaporator (MEE) is applied.

**Figure 2 molecules-30-00602-f002:**
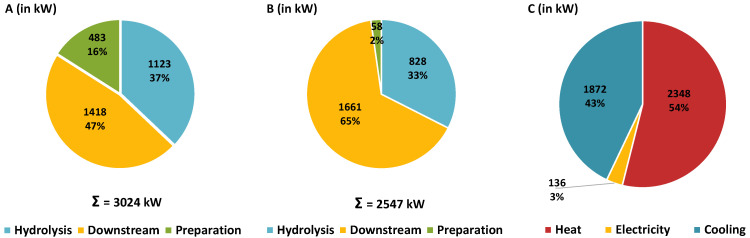
Required installed power for the 30 kt/a wheat straw dry mass reference case in Figure 1: (**A**) heating power of analyzed process areas before heat integration; (**B**) cooling power of the analyzed process areas before heat integration; (**C**) required installed power after heat integration.

**Figure 3 molecules-30-00602-f003:**
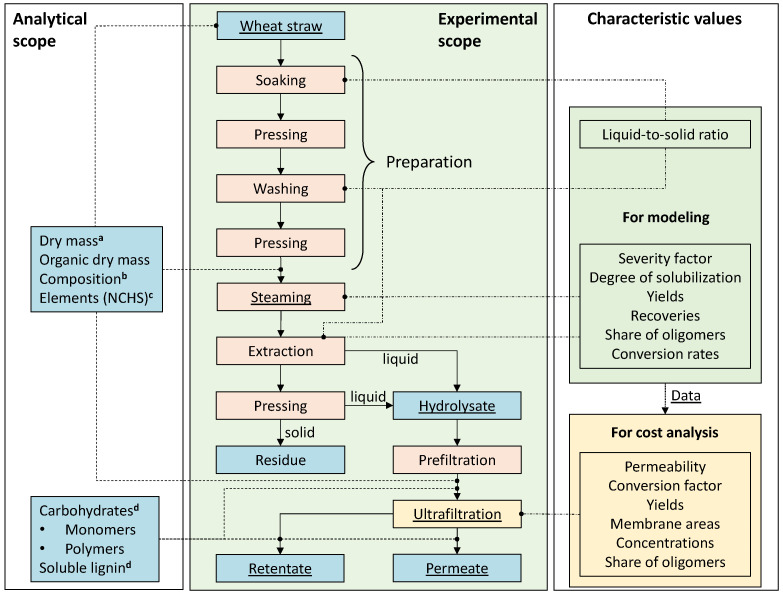
Scheme of the procedure for the experimental and analytical investigation. The associated defined characteristic values (for modeling see Section 3.2 and for cost analysis Section 3.3), process steps (orange) and analyzed fractions (blue) are differentiated by color. The main objects of investigation are underlined. Modeled process areas are indicated in green. ^a^ DIN EN ISO 18122; ^b^ NREL LAP TP-510-42618; ^c^ [47]; ^d^ NREL LAP TP-510-42618.

**Figure 4 molecules-30-00602-f004:**
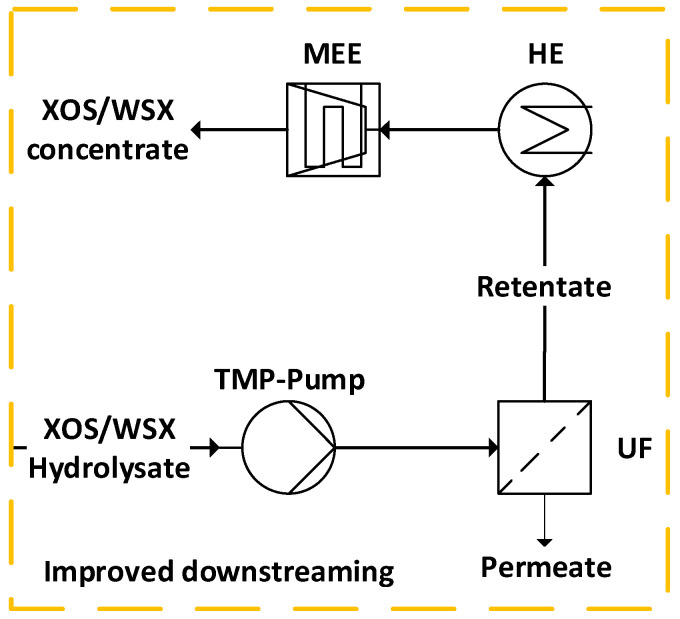
Simplified flowsheet of the improved downstream processing for implementation of an ultrafiltration (UF) step before the multi-effect evaporator (MEE) with a conversion factor of 50% (HE, heat exchanger; TMP, transmembrane pressure; WSX, water-soluble xylan; XOSs, xylooligosaccharides).

**Figure 5 molecules-30-00602-f005:**
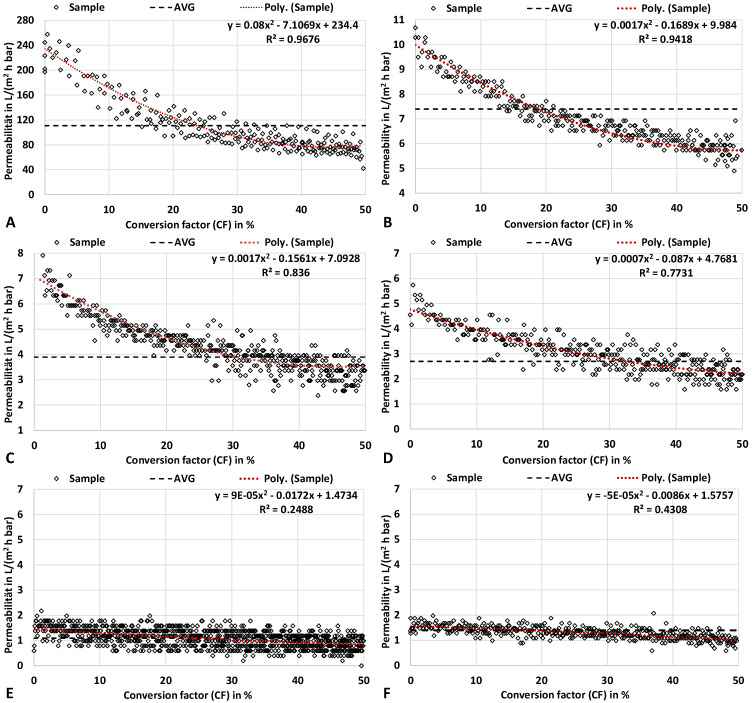
Course of permeability (rhombus) during filtration of the hydrolysate over the conversion factor. The dashed line indicates the (normalized) average permeability (*AVG*) and red dotted line the polynomial regression curve (Poly.) for sample filtration. (**A**): UH50 with 50 kDa MWCO at 1 bar *TMP*; Δ*t* = 5 s. (**B**): UH30 with 30 kDa MWCO at 4 bar *TMP*; Δ*t* = 10 s. (**C**): UP20 with 20 kDa MWCO at 4 bar *TMP*; Δ*t* = 10 s. (**D**): UP10 with 10 kDa MWCO at 4 bar *TMP*; Δ*t* = 10 s. (**E**): UP5 with 5 kDa MWCO at 4 bar *TMP*; Δ*t* = 10 s. (**F**): UH4 with 4 kDa MWCO at 4 bar *TMP*; Δ*t* = 20 s.

**Figure 6 molecules-30-00602-f006:**
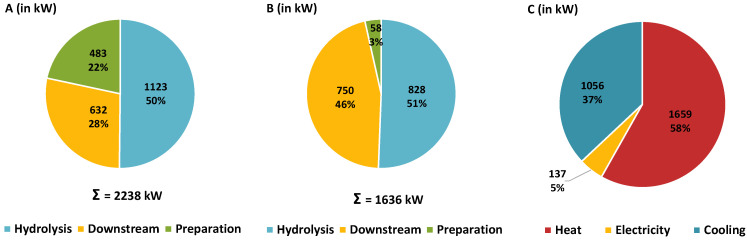
Required installed power for the 30 kt/a wheat straw dry mass alternative case with improved downstream processing: (**A**) heating power of the analyzed process areas before heat integration; (**B**) cooling power of the analyzed process areas before heat integration; (**C**) required installed power after heat integration.

**Figure 7 molecules-30-00602-f007:**
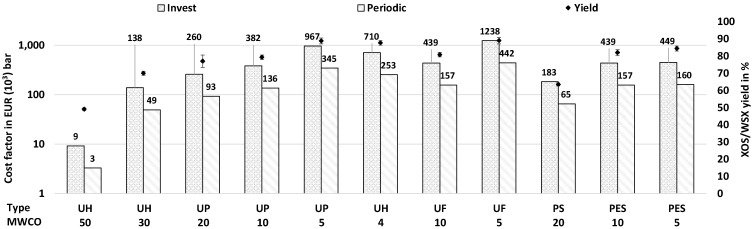
Calculated transmembrane pressure (*TMP*)-related cost factors (bars, left axis). Cost factors for investment (Equation (10)) and periodic renewal (Equation (11)) for a plant size of 30 kt_DM_/a wheat straw and the corresponding xylooligosaccharides (XOSs) / water-soluble xylan (WSX) yield (rhombi with standard deviation as error bar, right axis) in the retentate at a conversion factor of 50% (Equation (8)) for the investigated membranes (Table 3).

**Table 1 molecules-30-00602-t001:** Hydrolysate composition. Analytical results and respective standard deviations (STDs) according to [29] (DM dry mass; FM fresh mass; oDM organic dry mass; WS wheat straw; CB Cellobiose).

Fraction	DM	oDM	Glucan	Arabino-xylan	Lignin	Acetate	Ash	Protein	Rest
Unit	wt%_FM_	wt%_DM_	wt%_DM_	wt%_DM_	wt%_DM_	wt%_DM_	wt%_DM_	wt%_DM_	wt%_DM_
Hydrolysate	4.5	93.0	5.5	57.9	6.8	7.1	7.0	2.3	13.3
STD	0.1	0.3	0.2	1.2	0.1	0.2	0.3	0.3	1.3
**Component**	**CB**	**Glucose**	**Xylose**	**Arabinose**	**Formic Acid**	**Acetic Acid**	**HMF ^c^**	**Furfural**	**Ara/Xyl**
Unit	g/L	g/L	g/L	g/L	g/L	g/L	g/L	g/L	
Total concentrations ^b^									
Hydrolysate	0.82	1.75	26.11	2.06	0.80	3.28	0.11	0.92	7.9%
STD	0.01	0.06	0.08	0.02	0.00	0.02	0.01	0.03	0.07%
Released from oligomers									
Hydrolysate	0.82	1.67	22.75	0.62	0.00	0.87	0.01	0.42	2.7%
STD	0.01	0.06	0.07	0.01	0.00	0.02	0.00	0.03	0.06%
*S^ol^* (%) ^a^	--	95	87	30	0	26	8	46	--

^a^ Share of oligomers for the analyzed components in the hydrolysate, i.e., the share of the respective component in the hydrolysate that is released from non-monomers during analytical hydrolysis (for xylose, e.g., 22.75/26.11 = 0.87); ^b^ measured using HPLC after analytical hydrolysis with H_2_SO_4_; ^c^ 5-Hydroxymethylfurfural.

**Table 2 molecules-30-00602-t002:** Energy- and product mass-related process data for the conventional process at a simulated scale of 30 kt/a wheat straw dry mass and a yearly operating time of 7500 h/a (full load) after heat integration (Figure 2C). Xylooligosaccharides (XOSs) and water-soluble xylan (WSX) are the target fractions.

	Unit	Conventional Process ^a^
Heating Preparation Hydrolysis Downstream processing	MWh/a (%) MWh/a (%) MWh/a (%) MWh/a (%)	17,610 (100) 0 (0) 8453 (48) 9157 (52)
High Pressure steam Low Pressure steam	MWh/a (%) t/a MWh/a (%) t/a	8075 (46) 17,070 9538 (54) 15,630
Cooling Preparation Hydrolysis Downstream prodessing	MWh/a (%) MWh/a (%) MWh/a (%) MWh/a (%)	14,040 (100) 421 (3) 1123 (8) 12,496 (89)
Electricity	MWh/a (%)	1020 (100)
XOS/WSX concentrate Dry mass (DM) XOS/WSX content	t/a wt% %_DM_	10,172 50 61
Solid residue Dry mass (DM) Cellulose Lignin	t/a wt% %_DM_ %_DM_	47,733 50 46 34

^a^ Minor rounding deviations from Figure 2C are possible.

**Table 3 molecules-30-00602-t003:** Evaluated membranes, including details on manufacturers and typical operating conditions, as provided by the distributors [29].

Membrane Type	MWCO ^a^ kDa	Material ^b^	Supplier	pH	*T* °C	*p* bar	Permeability ^c^ L/(m^2^ h bar)
UH050	50	Hydrophilic PES	Microdyn Nadir	0–14	5–95		≥ 85
UH030	30	Hydrophilic PES	Microdyn Nadir	0–14	5–95		≥ 35
UP020	20	PES	Microdyn Nadir	0–14	5–95		≥ 70
UP010	10	PES	Microdyn Nadir	0–14	5–95		≥ 50
UP005	5	PES	Microdyn Nadir	0–14	5–95		≥ 10
UH004P	4	Hydrophilic PES	Microdyn Nadir	0–14	5–95		≥ 7.0
UF10	10	PES	Microdyn Nadir	2–11	5–45	1–21	≥ 74
UF5	5	PES	Microdyn Nadir	2–11	5–45	1–21	≥ 8.3
PS (GR61PP)	20	PS	Alfa Laval	1–13	5–75	1–10	
PES (GR80PP)	10	PES	Alfa Laval	1–13	5–75	1–10	
PES (GR90PP)	5	PES	Alfa Laval	1–13	5–75	1–10	

^a^ MWCO, nominal molecular weight cut-off; ^b^ PES—Polyethersulfone, PS—Polysulfone; ^c^ for respective test conditions.

**Table 4 molecules-30-00602-t004:** Energy- and product mass-related process data for the improved process at a simulated scale of 30 kt/a wheat straw dry mass and a yearly operating time of 7500 h/a (full load) after heat integration (Figure 6C). Xylooligosaccharides (XOSs) and water-soluble xylan (WSX) are the target fraction.

	Unit	Improved Process	Change to Reference Case ^b^
Heating Preparation Hydrolysis Downstream processing	MWh/a (%) MWh/a (%) MWh/a (%) MWh/a (%)	12,443 (100) 0 (0) 8461 (68) 3982 (32)	−29.3% (+41.7%) (−38.5%)
High Pressure steam Low Pressure steam	MWh/a (%) t/a MWh/a (%) t/a	8075 (65) 17,070 4369 (35) 7163	0% 0% −54.2% −54.2%
Cooling Preparation Hydrolysis Downstream processing	MWh/a (%) MWh/a (%) MWh/a (%) MWh/a (%)	7920 (100) 396 (5) 1901 (24) 5623 (71)	−43.6% (+66.7%) (+200.0%) (−20.2%)
Electricity	MWh/a (%)	1028 (100)	+0.9%
XOS/WSX concentrate Dry mass (DM) XOS/WSX content	t/a wt% %_DM_	7698 ^a^ 50 71 ^a^	−24.3% 0% +16.4%
Solid residue Dry mass (DM) Cellulose Lignin	t/a wt% %_DM_ %_DM_	47,733 50 46 34	0% 0% 0% 0%

^a^ Example calculated with regard to a 4 kDa membrane like UH4; ^b^ change in relative values in brackets.

**Table 5 molecules-30-00602-t005:** Average permeabilities ($\bar{P}$), the specific membrane areas ($A_{m}^{sp})$ and expected concentrations of targets ^a^ in permeate ($c_{XOS/WSX}^{per}$) and retentate ($c_{XOS/WSX}^{ret}$) for a simulated plant of 30 kt/a wheat straw dry mass.

Type ^b^ MWCO	Unit	UH 50	UH 30	UP 20	UP 10	UP 5	UH 4	UF 10	UF 5	PS 20	PES 10	PES 5
$\bar{P}$	l/(m^2^ h bar)	111	7.4	3.9	2.7	1.1	1.4	2.3	0.8	5.6	2.3	2.3
$A_{m}^{sp}$ ^c^	m^2^ bar	66	986	1854	2725	6904	5069	3138	8840	1306	3138	3208
$c_{XOS/WSX}^{ret}$	g/L	48	68	75	77	87	85	79	87	62	80	82
$c_{XOS/WSX}^{per}$	g/L	44	30	25	21	9	13	17	9	33	16	16

^a^ Referred as the sum of the non-monomeric xylose (XOS/WSX) and arabinose for a conversion factor of 50% (Equation (7)); ^b^ Characterized in Table 3; ^c^ Specific scalable membrane area (Equation (9)).

## Data Availability

The raw data supporting the conclusions of this article will be made available by the authors on request.

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
