# Peer review of "Energy-Related Assessment of a Hemicellulose-First Concept—Debottlenecking of a Hydrothermal Wheat Straw Biorefinery"

_molecules, 2025, doi:10.3390/molecules30030602_

Round 1

Reviewer 1 Report

Comments and Suggestions for Authors

The authors provided a process analysis based on experiments reported in a previously published article and compared the processes with or without ultrafiltration. Sevel major comments are as follows:

1.     This work seems mainly on the comparison of different ultrafiltration methods following the published method. Did the authors consider the ultrafiltration step the bottleneck of the whole conversion process?

2.     The authors focused on discussing the hemicellulose-first approach, aiming to produce xylan-derived hydrolysates, including XOS and water-soluble xylan. However, the whole utilization of lignocellulose is of the utmost importance when it comes to practical application. Without consideration of the influence of the treatment methods on the utilization of cellulose and lignin components, the analysis may be of less importance and provide less useful information.

3.     The cost analysis was only mentioned in Figure 7 on the process of ultrafiltration. Again, it seems the research content does not fit the title of the article.

4.     Why did the authors choose wheat straw as the substrate to represent the hemicellulose-first conversion approach? Corncob and corn stalk have higher hemicellulose content or larger amounts compared to wheat straw.

5.     In the overall discussion, the authors mentioned the current price of xylose molasses ranges from $0.1 to $0.3 per kg and XOS ranges from $10 to $30 per kg but did not compare with the technology reported in the article.

6.     The format of the article is unusual and not recommended, especially with discussion following each result, and provides an overall discussion. Besides, the separated items as shown in lines 101-110 are also not recommended. 

Author Response

Comment 1: This work seems mainly on the comparison of different ultrafiltration methods following the published method. Did the authors consider the ultrafiltration step the bottleneck of the whole conversion process?

Response 1: Thank you for your time and effort that serve in improving the work. That is indeed an important point to clarify. In general, the bottlenecks of lignocellulosic biorefineries are complex processes and high production costs - mainly due to extensive pretreatment and complex downstream processing. If a hydrothermal fractionation strategy is to be implemented, the high energy requirement and complex product purification are typically the cost drivers (see introduction second and the following paragraphs). We investigate and evaluate a possible solution for these challenges. A hemicellulose-first concept can be implemented with a comparatively low energy requirement (section 2.2). However, the energy requirement in conventional purification via multi-effect evaporators is high (typical challenge hydrothermal concepts). Ultrafiltration can be one possible solution for this issue. However, this can only be implemented effectively if the pre-treatment is designed accordingly and the interactions between membranes and the medium are taken into account. This is examined in detail in section 3. This work is therefore by no means aimed at “comparison of different ultrafiltration methods”, but shows through this investigation that the result of ultrafiltration strongly depends on the boundary conditions and must be investigated in practice. In addition, we show a suitable methodology (from the raw material to the evaluation of performance) how this can be implemented in practice. Ultrafiltration can therefore (under suitable conditions) contribute to the debottlenecking of lignocellulosic biorefinery concepts. In the Introduction, this is indicated from line 85:

“The applied combination of adapted steam treatment optimized for a subsequent membrane based solvent separation can be realized under relatively mild conditions and with low technical complexity. Especially the membrane processes can help to overcome a typical bottleneck of such lignocellulose biorefineries – the high energy demand in the downstream processing, in particular for thermal energy [7,17,30]. Nevertheless, a major drawback of membrane processes are their relatively high costs especially for investment and periodic membrane replacement depending strongly on the respective operation conditions and the properties of the feed solution [31,32]. The energy demand and ultrafiltration requirements must therefore be examined in detail based on experimental results.”

Comment 2: The authors focused on discussing the hemicellulose-first approach, aiming to produce xylan-derived hydrolysates, including XOS and water-soluble xylan. However, the whole utilization of lignocellulose is of the utmost importance when it comes to practical application. Without consideration of the influence of the treatment methods on the utilization of cellulose and lignin components, the analysis may be of less importance and provide less useful information.

Response 2: Thank you for pointing this out. We completely agree with this comment. For this reason, the focus of this work is clearly defined in line 97 of the definition of objectives:

“Since the production of XOS/WSX must be linked to an established processing of cellulose and lignin for holistic feedstock utilization, only the process route for the hemicellulose fraction is examined in more detail.”

This is possible because the influence of autohydrolysis on cellulose and lignin has been well studied and there is a great deal of literature on the subject. We use saturated steam without additional auxiliaries, which means that the effects on cellulose and lignin do not differ significantly from hot water hydrolysis or steam explosion. However, due to the comparatively mild conditions (180°C, 30 - 35 min) and the absence of the explosion, the crystalline structure of the cellulose is disintegrated less effectively. The lignin is also less “washed out” than with hot water hydrolysis. Nevertheless, we found very good biodegradability of cellulose (presumably due to the very effective removal of hemicellulose) (Figure S4 in supplementary material). The lignin is typically partially thermally degraded via condensation and formation of pseudo-lignin (i.e., degradation of ether and ester bonds and formation of C-C bonds). All of this is known and there are possible applications for these products, but it is not the focus of this work; cellulose and lignin are referred as a solid residue for the energetic assessment and are also taken into account in the results (see Figure 1, Tables 2 and 4). More details on the composition of the solid residue are already presented in the experimental study (10.1016/j.biortech.2023.130071). Line 130 states in relation to this:

“The approach enables the complete utilization of the raw material wheat straw, whereby the cellulose and lignin fraction should remain as completely as possible in the solid residue and should not contaminate the hemicellulose hydrolysate [29]. Here, however, only the processing of the hemicellulose (i.e., arabinoxylan) faction is examined in more detail.”

The utilization and the added value of all fractions in wheat straw must be determined in detail using suitable methods as part of a techno-economic evaluation, which we will soon be working on. However, this is a separate study for which the foundations must first be laid - this will be done in this manuscript, among others. We also modified the introduction in line 50 to emphasize the holistic feedstock utilization:

“Furthermore, no general concept has yet been established that combines complete feedstock utilization, low operating costs and high yields of products possessing significant market value.”

Comment 3: The cost analysis was only mentioned in Figure 7 on the process of ultrafiltration. Again, it seems the research content does not fit the title of the article.

Response 3: Many thanks for the hint. Can you describe in more detail why the content and the title should not fit?

This has the following background: Since the success of ultrafiltration depends on many boundary conditions (as the results confirm), the energy savings must be compared with the necessary effort. For this purpose, we define a parameter with which the different membranes can be evaluated in the context of the objective (i.e., energy-related assessment). Since the implementation of ultrafiltration as an alternative to a multi-effect evaporator mainly incurs costs (investment and periodic costs), they must also be taken into account accordingly. However, the cost analysis is only carried out for the derivation of a representative and transferable assessment parameter that takes into account the specific membrane area, permeability and transmembrane pressure (section 3.3). Of course, these results provide an excellent basis for further studies such as full techno-economic evaluations and comparisons, as further (direct and indirect) costs can now be easily taken into account.

We have now adapted the following text from line 321:

“In the following, the investigation of the improved process (alternative case) is described. Therefore, an extension of the process within the downstream processing by including ultrafiltration is investigated. The energy requirements and the equipment costs for the membrane modules are examined in detail, as they must be compared with the potential energy savings. They are also directly dependent on the hydrolysate filtration properties and therefore provide a quantifiable criterion of the adapted saturated steam pretreatment [29].”

And further in line 374 is stated:

“The equipment costs for ultrafiltration modules are determined reflecting the differences of the investigated membranes. Therefore, only the equipment costs are used; other costs are assumed to depend linearly on these equipment costs and thus not considered here.”

The results show how important it is to adequately assess an ultrafiltration step (as the results vary considerably). Because the assessment methodology must be transparent and reproducible, we do this in detail (this is often neglected in purely techno-economic studies).

Comment 4: Why did the authors choose wheat straw as the substrate to represent the hemicellulose-first conversion approach? Corncob and corn stalk have higher hemicellulose content or larger amounts compared to wheat straw.

Response 4: That is indeed an important question. Corn cobs and corn straw have a higher hemicellulose content and are also easier to process feedstock for other reasons. This is also one of the reasons why it is important to develop more effective processes for residues that have not yet been fully utilized. There is considerable competition for corn cobs (e.g. feed market), while some cereal straw is still burned in the fields. In addition, wheat straw is available worldwide in large quantities as a byproduct (and is relatively similar in composition to other types of cereal straw), while corn cobs are only produced in regions with intensive maize cultivation. Due to its wide availability and low competitive use, wheat straw tends to be cheaper as a raw material. Despite technical challenges, its versatile chemical composition provides a robust basis for the production of bioenergy, biofuels and bio-based products (backed by literature; see introduction). Our aim is therefore not to design the most productive concepts for hemicellulose (XOS/WSX) production, but to investigate more effective utilization concepts for wheat straw (as an example) that is difficult to utilize (in multi-product high-value biorefineries). This was also mentioned in the introduction in the first and fourth paragraph. These have now been adapted to emphasize the reasons for wheat straw.

“Next-generation biorefineries aim to increasingly utilize lignocellulosic biomass. Agricultural residues in particular, such as wheat straw, are available worldwide, are underutilized and are not in direct competition with the food and animal feed market [1].”

“Wheat straw, as a well-researched example of a lignocellulosic feedstock, has significant potential for the provision of biofuels and bio products [11,12]. As an agricultural residue (sometimes without direct competitive use), it can be a comparatively inexpensive feedstock [12]. Nevertheless, it is still challenging to exploit this potential in biorefineries on an industrial scale under commercial conditions [4,13].”

Comment 5: In the overall discussion, the authors mentioned the current price of xylose molasses ranges from $0.1 to $0.3 per kg and XOS ranges from $10 to $30 per kg but did not compare with the technology reported in the article.

Response 5: Many thanks for the comment. This information is merely intended to clarify the motivation for the effective valorization of hemicellulose. Hemicellulose is still given little attention in many biorefinery concepts, but we see considerable potential in it. This is simply to show how much potential there is for further value creation in pentose-based hemicellulose and why a detailed examination of this fraction makes sense. However, we agree that a comparison of the production costs with the proposed concept must be made. This will be done in detail in another (techno-economic) study. We have revised the paragraph (line 621) to make the point clear:

“This study evaluates a hydrothermal hemicellulose-first concept for wheat straw biorefineries, focusing on the hemicellulose valorization. A more effective use of hemicellulose, such as producing xylooligosaccharides (XOS) and water-soluble xylan (WSX), offers significant opportunities for higher-value applications compared to e.g., xylose molasses. Notably, while molasses typically sells for $0.1 to $0.3 per kg, XOS can command prices ranging from $10 to $30 per kg [27,58]. However, for practical implementation in biorefineries, the results must be considered in the broader context of overall lignocellulose fractionation and valorization. The concept investigated here demonstrates energy-efficient production of XOS/WSX concentrates (characterized in Table 4 and Figures S1 and S2) under optimized conditions (section 3.4.2). The integration of the concept into existing hydrothermal biorefineries can contribute to the overall economic viability of multi-product wheat straw biorefineries. “

If this information is confusing or misleading, we are welcome to remove it.

Comment 6: The format of the article is unusual and not recommended, especially with discussion following each result, and provides an overall discussion. Besides, the separated items as shown in lines 101-110 are also not recommended.

Response 6: We are sorry that the selected format is not satisfactory. Can you clarify what the specific problem is?

In lines 101-110, bullet points are used to define the two equally important objectives of the study. This is quite common in scientific journals. The format of the paper is determined by the fact that in section 2 we model and simulate a process that is fully described in another experimental study (10.1016/j.biortech.2023.130071). Section 3 then examines an improved process, for which the experimental procedure, research methodology and key figures must also be described. As a result, they cannot be discussed at the same level of content - this would be scientifically inconsistent. Our solution is a compromise indeed (and certainly unusual), but in our view it supports the content and helps the reader to gain an overview and to understand the content.

There are always pro and con arguments to mix or separate results and discussion. The results are straightforward, while the discussion and interpretation are debatable. We separate them to ensure a better overview. Otherwise, it is often difficult to understand where the interpretation of the results begins in complex and extensive studies. This serves the clarity of the content. The detailed results are discussed directly afterwards, so that the data and interpretation are separated but the connection is still clear. From our point of view, it is always advisable and scientifically sound to separate results and data from interpretation and discussion. Especially in the field of natural sciences and engineering.

The overall discussion deals with the benefits of the results in the context of lignocellulose utilization in biorefineries, and therefore takes place at a different level than the detailed results of sections 2.2 and 3.4. Since it includes and evaluates the results of sections 2 (reference process) and 3 (improved process), it must be at the same level structurally (i.e., section 4).

Further small changes and corrections have been made to the manuscript and are marked in red.

Reviewer 2 Report

Comments and Suggestions for Authors

This work investigated the hot water extraction of WSX and XOS from wheat straw to isolate this fraction from the cellulose and lignin in the wheat straw. The WSX and XOS are then purified as a value-added co-product, while the cellulose and hemicellulose can be valorised separately. The focus of the study was on energy requirements and improvements. The study is well-designed and comprehensive and can add value to the overall field of cellulosic ethanol production, where xylan and XOS is not adequately valorised which impacts the overall economics of the process. This investigation has substantial merit.

While the rationale for limiting the scope of the study (to extraction and processing of the WSX and XOS) is sound, the overall success of a commercial process will rely on the valorisation of the entire feedstock, including primary cellulose and lignin components. Therefore, integrating this process with the rest of the biorefinery will be critical. However, this does not detract from the value of the study at all.

Author Response

Comment 1: This work investigated the hot water extraction of WSX and XOS from wheat straw to isolate this fraction from the cellulose and lignin in the wheat straw. The WSX and XOS are then purified as a value-added co-product, while the cellulose and hemicellulose can be valorised separately. The focus of the study was on energy requirements and improvements. The study is well-designed and comprehensive and can add value to the overall field of cellulosic ethanol production, where xylan and XOS is not adequately valorised which impacts the overall economics of the process. This investigation has substantial merit.

Response 1: Thank you for your time and effort that serve in improving the work. It is gratifying that the aim and content of the work have been prepared in an understandable way.

Comment 2: While the rationale for limiting the scope of the study (to extraction and processing of the WSX and XOS) is sound, the overall success of a commercial process will rely on the valorisation of the entire feedstock, including primary cellulose and lignin components. Therefore, integrating this process with the rest of the biorefinery will be critical. However, this does not detract from the value of the study at all.

Response 2: Thank you for pointing this out. We completely agree with this comment. For this reason, the focus of this work is clearly defined in line 97 of the definition of objectives:

“Since the production of XOS/WSX must be linked to an established processing of cellulose and lignin for holistic feedstock utilization, only the process route for the hemicellulose fraction is examined in more detail.”

This is possible because the influence of autohydrolysis on cellulose and lignin has been well studied and there is a great deal of literature on the subject. We use saturated steam without additional auxiliaries, which means that the effects on cellulose and lignin do not differ significantly from hot water hydrolysis or steam explosion. However, due to the comparatively mild conditions (180°C, 30 - 35 min) and the absence of the explosion, the crystalline structure of the cellulose is disintegrated less effectively. The lignin is also less “washed out” than with hot water hydrolysis. Nevertheless, we found very good biodegradability of cellulose (presumably due to the very effective removal of hemicellulose) (Figure S4 in supplementary material). The lignin is typically partially thermally degraded via condensation and formation of pseudo-lignin (i.e., degradation of ether and ester bonds and formation of C-C bonds). All of this is known and there are possible applications for these products, but it is not the focus of this work; cellulose and lignin are referred as a solid residue for the energetic assessment and are also taken into account in the results (see Figure 1, Tables 2 and 4). More details on the composition of the solid residue are already presented in the experimental study (10.1016/j.biortech.2023.130071). Line 130 states in relation to this:

“The approach enables the complete utilization of the raw material wheat straw, whereby the cellulose and lignin fraction should remain as completely as possible in the solid residue and should not contaminate the hemicellulose hydrolysate [29]. Here, however, only the processing of the hemicellulose (i.e., arabinoxylan) faction is examined in more detail.”

The utilization and the added value of all fractions in wheat straw must be determined in detail using suitable methods as part of a techno-economic evaluation, which we will soon be working on. However, this is a separate study for which the foundations must first be laid - this will be done in this manuscript, among others. We also modified the introduction in line 50 to emphasize the holistic feedstock utilization:

“Furthermore, no general concept has yet been established that combines complete feedstock utilization, low operating costs and high yields of products possessing significant market value.”

Further small changes and corrections have been made to the manuscript and are marked in red.
